# The Pathophysiology of Long COVID throughout the Renin-Angiotensin System

**DOI:** 10.3390/molecules27092903

**Published:** 2022-05-02

**Authors:** Shaymaa Khazaal, Julien Harb, Mohamad Rima, Cédric Annweiler, Yingliang Wu, Zhijian Cao, Ziad Abi Khattar, Christian Legros, Hervé Kovacic, Ziad Fajloun, Jean-Marc Sabatier

**Affiliations:** 1Faculty of Sciences 3, Department of Biology, Lebanese University, Campus Michel Slayman Ras Maska, Tripoli P.O. Box 45061, Lebanon; shaymaa.khazaal@hotmail.com; 2Faculty of Medicine and Medical Sciences, University of Balamand, Dekouene Campus, Sin El Fil P.O. Box 55251, Lebanon; julienharb0408@gmail.com; 3Laboratory of Applied Biotechnology (LBA3B), Azm Center for Research in Biotechnology and Its Applications, EDST, Lebanese University, Tripoli P.O. Box 45061, Lebanon; mohamad.rima@hotmail.com; 4Department of Geriatric Medicine and Memory Clinic, Research Center on Autonomy and Longevity, University Hospital & Laboratoire de Psychologie des Pays de la Loire, LPPL EA 4638, SFR Confluences, University of Angers, 44312 Angers, France; cedric.annweiler@chu-angers.fr; 5State Key Laboratory of Virology, Modern Virology Research Center, College of Life Sciences, Wuhan University, Wuhan 430072, China; ylwu@whu.edu.cn (Y.W.); zjcao@whu.edu.cn (Z.C.); 6Laboratory of Georesources, Geosciences and Environment (L2GE), Microbiology/Tox-Ecotoxicology Team, Faculty of Sciences 2, Lebanese University, Campus Fanar, Jdeidet El-Matn, Beirut P.O. Box 90656, Lebanon; ziad.abikhattar@ul.edu.lb; 7INSERM, CNRS, MITOVASC, Team 2 CarMe, SFR ICAT, University of Angers, 49000, France; christian.legros@univ-angers.fr; 8Institut de Neurophysiopathologie (INP), Aix-Marseille Université CNRS, 13385 Marseille, France; herve.kovacic@univ-amu.fr

**Keywords:** SARS-CoV-2, COVID-19, Long COVID, renin-angiotensin system, angiotensin-converting enzyme 2, Angiotensin II receptor type 1, Angiotensin II receptor type 2, infectious disease, immunity disorders

## Abstract

COVID-19 has expanded across the world since its discovery in Wuhan (China) and has had a significant impact on people’s lives and health. Long COVID is a term coined by the World Health Organization (WHO) to describe a variety of persistent symptoms after acute SARS-CoV-2 infection. Long COVID has been demonstrated to affect various SARS-CoV-2-infected persons, independently of the acute disease severity. The symptoms of long COVID, like acute COVID-19, consist in the set of damage to various organs and systems such as the respiratory, cardiovascular, neurological, endocrine, urinary, and immune systems. Fatigue, dyspnea, cardiac abnormalities, cognitive and attention impairments, sleep disturbances, post-traumatic stress disorder, muscle pain, concentration problems, and headache were all reported as symptoms of long COVID. At the molecular level, the renin-angiotensin system (RAS) is heavily involved in the pathogenesis of this illness, much as it is in the acute phase of the viral infection. In this review, we summarize the impact of long COVID on several organs and tissues, with a special focus on the significance of the RAS in the disease pathogenesis. Long COVID risk factors and potential therapy approaches are also explored.

## 1. Introduction

Severe acute respiratory syndrome coronavirus 2 (SARS-CoV-2) infection (also called COVID-19) resulted in a worldwide pandemic with substantial mortality and morbidity. Approximately 80% of infected patients developed mild to moderate disease, and 5% of those with severe disease developed critical illness [1]. The recovery from a mild SARS-CoV-2 infection takes 7–10 days from the onset of symptoms, but severe sickness recovery can take between three to six months. A substantial percentage of patients who recovered from COVID-19 have one or more persistent, acute COVID-19 symptoms or new symptoms that last for weeks or even months [2]. The manifestation of these symptoms can be continuous, relapsing, remitting, and occur irrespective of the viral status [3]. These long-lasting complications have been called “post COVID syndrome” or “long COVID”, and the people with it “long haulers” [4]. In fact, long COVID is divided into two stages: the “post-acute COVID” where symptoms last for more than three weeks, and “chronic COVID” where symptoms stay more than 12 weeks [5]. Long COVID symptoms include fatigue, headache, and upper respiratory tract complaints (i.e., dyspnea, sore throat, persistent cough, and smell loss). Multi-system complaints were also reported as long COVID symptoms including ongoing fever and gastroenterological symptoms [6]. According to a recent meta-analysis, the five most prevalent signs of long COVID were fatigue (58%) headache (44%), attention deficit (27%), hair loss (25%), and dyspnea (24%) [7]. Skin rashes, palpitations, diarrhea, and a ‘pins and needles’ sensation were among the additional symptoms reported. The complications were not only observed at the physiological level, but also psychological complications such as anxiety, depression, and post-traumatic stress disorder (PTSD) were linked to long COVID. The first symptoms of long COVID appeared three to four weeks after acute symptom onset [6], and symptoms persisted even after 60 days [8]. As such, COVID-19 patients, who were discharged from the hospital, had shortness of breath and were extremely tired even after three months [9].

The main risk factors of long COVID include sex, age, and the health of the patient. In fact, women are twice as likely as males to get long COVID [3]. In addition, patients with long COVID are reported to be around four years older than those who do not have long COVID [3,10]. The presence of comorbidities also increased the risk for developing long COVID. The number of symptoms present during the acute stage of disease were also suggested as a risk factor for long COVID. As such, having more than five symptoms was linked to a higher chance of developing long COVID [6]. This did not exclude the possibility that even individuals showing minor symptoms at first were found to have developed long COVID [10]. 

In this review, we highlight the impact of long COVID on several systems, including the central nervous, cardiovascular, respiratory, immune, and urinary systems as well as with regard to diabetes (Table 1). We also discuss the pathophysiological relationship between long COVID and the renin angiotensin system (RAS). Long COVID potential therapeutic approaches will also be reviewed. 

## 2. The Interplay between Long COVID and Renin-Angiotensin System (RAS)

SARS-CoV-2 infects cells through attaching to the angiotensin-converting enzyme-2 (ACE2) and so directly impacts the RAS (for review, see [11]). This hormonal and enzymatic system regulates not only cardiovascular homeostasis [12], but also pulmonary, renal, and innate immunological systems, as well as the gut microbiome [13,14].

At the molecular level, the renin released by the kidneys reduces angiotensinogen secreted by the liver to create angiotensin I (Ang I) (Figure 1A). Then, Ang I is cleaved by the angiotensin-converting enzyme (ACE) to give angiotensin II (Ang II) that is the ligand of the ACE2 receptor. Ang II binds to the angiotensin type-1 receptor (AT1R), leading to detrimental outcomes such as vasoconstriction, fibrosis, apoptosis, oxidative stress, and inflammation. However, by binding to angiotensin type-2 receptor (AT2R), Ang II promotes protective effects such as vasodilation, anti-apoptosis, anti-fibrosis, anti-oxidative, and anti-inflammatory effects. Moreover, Ang II can be cleaved by ACE2 to give angiotensin (1–7), which interacts with the G-protein-coupled receptor (GPCR) Mas (MasR), leading to protective effects similar to those mediated by AT2R [14,15]. Consequently, the RAS has two axes: (1) the proinflammatory ACE/Ang II/AT1R axis and (2) the anti-inflammatory ACE2/Ang-(1-7)/AT2R axis. The ACE2 receptor plays a major role in the equilibrium between the two axes, assuring RAS homeostasis.

Over-activation of the RAS has been identified as a critical factor in COVID-19 symptoms and pathologies [12]. In fact, SARS-CoV-2 activates the RAS by attaching, via its spike protein, to the ACE2 receptor, which is involved in the degradation of Ang II [16]. As a result, SARS-CoV-2 binding to ACE2 prevents the cleavage of Ang II (Figure 1B). The AT1R is, therefore, over-activated by the excess Ang II. The over-activated AT1R results in vasoconstriction, hypertension, inflammation, oxidative stress, heart hypertrophy, tissue fibrosis (heart, lungs, kidneys, and liver), ageusia (loss of taste), anosmia (loss of smell), neurological dysfunctions, obesity and diabetes, and lesions in the skin [12,15,17,18]. Therefore, this over-activation of AT1R is detrimental to the human body and could explain the relevance on long term complications of SARS-CoV-2 infection. The ubiquitous expression of ACE2 in different tissues multiplies the chances of RAS impairment caused by SARS-CoV-2 infection and underlies the virus pleiotropism [19]. 

The imbalance of RAS in COVID-19, via the virus-mediated down-regulation of ACE2, favorizes the pro-inflammatory ACE/Ang II/AT1R axis leading to the cytokine storm syndrome and consequent cellular damage. In fact, pro-inflammatory cytokine over-production, such as interleukin-1β (IL-1β), interleukin-6 (IL-6), interleukin-12 (IL-12), and interferon gamma (IFN-γ), results in the severe clinical consequences of SARS-CoV-2 infection over time, including pulmonary inflammation and significant lung damage [20,21]. The elevated levels of pro-inflammatory cytokines in the plasma aggravate systemic inflammation and lead to acute respiratory distress syndrome (ARDS), multi-organ failure, and death [11,20,22]. Therefore, it is likely that a direct link exists between RAS dysfunction and long COVID development. In addition, since serum ACE2 activity correlates with the severity of the infection and the mortality rates [23], it is possible that RAS is significantly impaired following severe infection, without excluding the possibility that mild or asymptomatic infections could also lead to long COVID via other signaling pathways. Clearly, COVID-19 is not a simple viral pneumonia, but rather one with pathophysiological consequences. 

## 3. Long COVID in the Nervous System 

### 3.1. The Central Nervous System

The persisting sequelae after SARS-CoV-2 infection involving the central nervous system (CNS) included predominant neurologic and psychiatric symptoms, such as memory and attention deficits, reduced capacity to do daily tasks, frequent headaches, changes in cutaneous sensation, autonomic dysfunction, chronic fatigue, and, in severe cases, delusions and paranoia [17,24,25]. Surprisingly, many patients showing these neurologic symptoms for more than a year after contracting COVID-19 are under 50 years old and were healthy and active before getting infected [26]. This supports the idea that the virus can have a serious impact on the CNS. In fact, imaging results of the UK Biobank cohort indicated specific areas of brain atrophy in COVID-19 patients, compared to a control group without COVID-19, highlighting a direct impact of the virus on the CNS [26]. Also, neuronal atrophy and degeneration of cranial nerves, including the olfactory nerve and the neighboring olfactory bulb, have previously been reported in patients with persistent hyposmia or anosmia after acute COVID-19 [27]. In agreement with these findings, patients with long COVID exhibited lower metabolic activity in their brain [26]. The etiology underlying these symptoms and alterations in the brain, however, remains unknown. Of note, affected zones, including the cerebral cortex and hippocampus, possess different members of the RAS system.

### 3.2. The Peripheral Nervous System

The peripheral nervous system (PNS) can be also affected by SARS-CoV-2 infection and show persistent symptoms for weeks or even months after the infection. Peripheral neuropathy symptoms differ depending on which nerves are damaged: motor, sensory, or autonomic. COVID-19 infection is thought to affect the autonomic nervous system (ANS). In fact, the well-known COVID-19 cytokine response storm is caused by sympathetic activity which causes pro-inflammatory cytokine release. Vagus nerve stimulation mediating anti-inflammatory responses highlight the relationship between SARS-CoV-2 infection and the PNS and suggest the ANS as a therapeutic target. Of note, vagus nerve dysfunction has also been reported in SARS-CoV-2 infection and proposed as a key pathophysiological hallmark of long COVID [28].

COVID-19-related autonomic dysfunction might be caused by the virus itself [29]. In fact, cohort studies have identified viral infections as a common precursor to autonomic diseases such as orthostatic hypotension and postural orthostatic tachycardia syndrome. These immune-mediated neurological disorders are caused by autoantibodies, such as those against α/β adrenoceptors and muscarinic receptors [30,31,32,33,34]. As a result, it could be possible that SARS-CoV-2 infection promotes autonomic dysfunctions through an autoimmune-mediated process. In agreement with this hypothesis, Guillain-Barré syndrome (GBS), a rare, autoimmune disorder that targets nerves, was associated with SARS-CoV-2 infection [35,36]. These effects are described in macrophage activation syndrome induced by RAS dysfunction, which is known to cause an alteration of the innate immunity responsible for autoimmune diseases.

Symptoms of fever, cough, interstitial pneumonia, hypo-ageusia, and hypo-anosmia were associated with GBS in conjunction with COVID-19 [37]. This hypothesis was also investigated by checking whether some symptoms of long COVID could be linked to a virus- or immune-mediated disturbance of the ANS, resulting in acute or long-term orthostatic intolerance syndromes. Orthostatic hypotension (OH), vasovagal syncope (VVS), and postural orthostatic tachycardia syndrome (POTS) are all orthostatic intolerance syndromes [38]. An aberrant autonomic reaction to orthostasis is central to the pathogenesis (standing up). In fact, blood accumulates in the pelvis and legs when a healthy individual stands, decreasing venous return. Baroreceptors in the heart and aorta perceive this and respond by raising sympathetic neuronal and adrenergic tone (mediated by norepinephrine and epinephrine, respectively). This causes tachycardia followed by splanchnic vascular bed vasoconstriction, which enhances venous return to the heart [38]. The release of adrenaline and norepinephrine in orthostatic intolerance induces the significant tachycardia, which manifests as palpitations, dyspnea, and chest pain, which are typical symptoms of long COVID [39]. The poor control of heart rate variability (HRV) in long COVID patients also supports the autonomic dysfunction in these patients. This dysregulation was linked to long COVID symptoms such as fatigue and hypoxia [40]. Signs of peripheral nerve and muscular system dysfunction were also reported after SARS-CoV-2 infection. The persistent symptoms of myalgia, weakness (or exercise intolerance), sensory dysfunctions (primarily positive symptoms, such as paraesthesia and neuropathic pain), and dysautonomia were all linked to long COVID [24]. 

### 3.3. The Outcomes of Neuroinflammation

An important role of neuroinflammation has also been linked to long COVID. In fact, abnormal humoral and cellular immune responses, systemic inflammatory markers such as IL-6, and autoantibodies targeting cellular receptors could all be involved in systemic and neurological long COVID sequelae [37]. The full scope of long COVID neurological problems has yet to be determined. Neuroinflammation and neuronal injury observed in acute COVID-19 raise the possibility that infection could speed up or induce the development of neurodegenerative illnesses like Alzheimer’s or Parkinson’s. There is currently no evidence on the neurodevelopmental trajectories of children who typically have modest COVID-19 and show low neurologic or neuropsychiatric symptoms during or after acute illness. Because of broad endothelial activation, which commonly involves the brain, those who suffer from the unusual multisystem inflammatory syndrome in children (MIS-C) may be at a higher risk for neurological complications [26]. 

### 3.4. SARS-CoV-2 Entry into the CNS and PNS

Hematogenous or transsynaptic pathways involving the ACE2 receptor, which is found on the surface of a variety of cells, including neurons, astrocytes, endothelial, and smooth muscle cells of cerebral blood vessels, and skeletal muscle cells, facilitate SARS-CoV-2 cell entry into the CNS and PNS [41]. In the CNS, ACE2 receptors are predominantly expressed in the olfactory bulb, amygdala, hippocampus, middle temporal gyrus, posterior cingulate cortex, and the brainstem; therefore, hyposmia, mood disorders, cognitive impairment, sleep disorders, and dysautonomia have been linked to the dysfunction of these ‘ACE2-rich’ brain areas [17,40,42,43]. SARS-CoV-2 could indeed infect brain cells such as neurons, astrocytes, and microglial cells (and possibly oligodendrocytes) by binding to their ACE2 receptors and, as a result, after entering the ear-nose-throat (ENT) sphere, attacking the olfactory bulb on the floor of the cranial box through the olfactory epithelium just below the nasal cavity level. The involvement of the brainstem and cerebellum is critical evidence suggesting that these brain regions may be involved in several neurological manifestations of long COVID which are similar to myalgic encephalomyelitis or chronic fatigue syndrome (ME/CFS) and POTS [27].

### 3.5. The Complexity of RAS in the CNS and Its Impairing in Long COVID 

Besides the endocrine RAS regulating water and electrolyte balance, it was demonstrated that neurons have intracrine and local types of RAS [44,45]. RAS regulates brain homeostasis primarily by the action of four angiotensin receptor subtypes: AT1R, AT2R, MasR, and angiotensin II type-4 receptor (AT4R). AT1R induces vasoconstriction, proliferation, inflammation, and oxidative stress, whereas AT2R and MasR mitigate AT1R’s actions. AT1Rs are divided into two types: AT1A and AT1B. The AT1A receptor is mostly found in brain regions that contribute to blood pressure and electrolyte balance homeostasis, whereas the AT1B receptor is found in structures like the cerebral cortex and hippocampus that are involved in memory and higher brain functions. AT4R regulates the release of dopamine and acetylcholine, as well as memory and learning consolidation [11,46]. AT2R is abundantly expressed in the brain. It plays a role in brain damage healing (i.e., axonal regeneration, neurite development) as well as inflammation reduction and vasodilation. In other words, AT2R has a neuroprotective function by helping neuronal survival and protecting the brain from damages. In this context, the AT2R protects against the harmful effects of AT1R activation, which occurs during SARS-CoV-2 infection [47,48]. It has been shown that SARS-CoV-2 has a brain tropism, and the neurological dysfunctions reported could be due to the impairment of RAS in the CNS (Figure 2). It has been shown that SARS-CoV-2 can infect nerve cells (such as neurons and astrocytes that express the ACE2 receptor) in vitro and in vivo [11,45,49,50,51,52]. The fact that SARS-CoV-2 impairs RAS pathways suggests that many nervous sequelae of long COVID could be explained by ACE2 receptor loss and AngII/AT1R over-activation. The virus binds to ACE2, resulting in its downregulation and a shift in the dynamic balance between the two faces of the RAS: (1) ACE/Ang II/AT1R, which has proinflammatory features, and (2) ACE2/Ang-(1-7)/AT2R, which has anti-inflammatory properties. This imbalance due to the overactivation of the Ang II/AT1R axis in the brain leads to hypertension, neuroinflammation, increased oxidative stress, BBB disruption, and neurotoxicity [11,42,46].

The abundancy of RAS in the brain and its involvement in different physiological and cognitive processes explain the broad consequences of SARS-CoV-2 infection. The diversity of the RAS, notably via its two types, circulating and local RAS, emphasizes its important role in homeostasis. As such, circulating RAS affects nuclei in the hypothalamus and medulla via circumventricular organs, which are brain areas that lack a blood-brain barrier (BBB), and transmit to nuclei in the hypothalamus and medulla, while the independent local RAS of the brain, b-RAS, generates all components of the circulatory RAS [44]. On the other hand, the prorenin receptor (PRR) is widely expressed in neurons, and some microglial cells of many vascular brain regions as well as the brain cortex and basal ganglia [53]. Overstimulation of this system activates the Ang II/AT1R axis and thus may lead to cognitive impairment. Furthermore, PRR can establish its own signaling pathway and produce pro-oxidative effects. Although a link connecting PRR and neural development has been shown, the RAS-independent functions of PRR in the brain are yet unknown [17,54]. 

AT1R signaling can be also triggered by ACE overexpression [55] and results in (i) second messenger signaling including inositol trisphosphate, diacylglycerol, and arachidonic acid, as well as the (ii) activation of downstream effectors including phospholipases C, A, and D after G-protein coupling stimulation of AT1R by Ang II. The AT1R signaling cascade activates protein kinase C, Akt, intracellular protein kinases and serine/threonine kinases (such as mitogen-activated protein kinase (MAPK) family kinases). These facts highlight the diversity of the components that can be touched by RAS impairment and their consequent sequelae. For example, hypertrophy, vascular remodeling, and hyperplasia may arise from overactivation of the AT1R cascade [11,55,56]. The Ang II/AT2Rs counteract AT1R’s effects by promoting phospholipase A2 and activating multiple protein phosphatases as well as the nitric oxide (NO)/cyclic GMP pathway, triggering the release of arachidonic acid [25,48]. AT2R also suppresses cell growth and proliferation by blocking insulin and EGFR autophosphorylation. Furthermore, by inhibiting negative feedback, AT1R blockage promotes angiotensinogen and AT2R stimulation [25,48]. This balance between AT1R and AT2R is important for maintaining a normal physiological environment. Therefore, its perturbation, specifically via the AngII/AT1R connection, results in a slew of potentially harmful consequences on the endothelium, inflammation, and coagulation, in addition to the well-known vasoconstrictive effects in the brain that may potentiate the post-acute COVID symptoms. Since the interactions between AngII and AT2R and Ang II and MasR operate as a vital “protective arm” to balance these effects [57], ACE2 is considered as a key component of the anti-RAS system in the genesis and prevention of illness. 

Positron emission tomography (PET) of long COVID patients, with persistent complaints at least three weeks after the onset of symptoms of acute infection, showed hypometabolism in their bilateral rectal/orbital gyrus (containing the olfactory gyrus), right temporal lobe (amygdala and hippocampus extending to the right thalamus), bilateral pons/medulla brainstem, and bilateral cerebellum. Interestingly, the metabolism of the frontal cluster (containing the olfactory gyrus) was worse in subjects receiving ACE inhibitors for high blood pressure, but better in subjects who received nasal decongestant spray [58]. These findings could point to the involvement of the ACE2 receptor in SARS-CoV-2 neurotropism, particularly in the olfactory bulb. This is likely due to the propagation pathway from the nose to the olfactory bulb, where the ACE2 receptor is strongly expressed, which has been hypothesized for various coronaviruses [59]. Cerebellar hypometabolism results in several symptoms including hyposmia/anosmia, memory/cognitive impairment, which is concordant with the involvement of this region in executive functions and working memory [60,61]. In addition, hypometabolism in the frontal cortex, brainstem, and cerebellum is associated with pain symptoms [62], particularly in fibromyalgia patients [63]. Other disorders have been linked to hypometabolism of the brainstem and cerebellum, such as insomnia and dysautonomia [64]. Taken together, the hypometabolism detected in brain regions of long COVID patients explain some of the persistent symptoms. The relationship between PET hypometabolism and the duration after the onset of the acute infection symptoms could suggest a link with the clinical severity. A more severe clinical profile is associated with a more severe PET hypometabolism and a longer duration of symptoms. Therefore, tracking functional brain activity can serve as a cerebral biomarker tool to classify long COVID patients regarding the severity of their symptoms and to distinguish affected patients from healthy subjects [58]. Also, these findings give insight to act quickly and restore brain metabolism in these patients before more severe diseases develop, such as Alzheimer’s disease (AD), which involves the severe reduction of the cerebral metabolic rate for glucose [65]. 

## 4. Evidence of RAS in the Cardiovascular System Highlight the Vulnerability of the System to Long COVID 

A cohort study was conducted to assess the risks and costs of pre-specified incidents of cardiovascular events in non-hospitalized, hospitalized, and intensive care unit patients of COVID-19. The study revealed that after one month of infection, people are more likely to have cardiovascular complications such as cerebrovascular disorders, dysrhythmias, ischemic and non-ischemic heart disease, pericarditis, myocarditis, heart failure, and thromboembolic illness. They were also at a significant risk of developing heart and pericardial inflammatory illness, as well as ischemic heart disease, such as coronary artery disease. Heart failure and cardiac arrest were also seen, as well as other cardiovascular disorders [66]. 

The relationship between COVID-19 and cardiovascular diseases development in long COVID need further investigation [67,68]. However, there are several probable mechanisms behind the fibrosis and cardiac tissue damage observed following SARS-CoV-2 infection. For example, direct viral entry into cardiomyocytes and subsequent cell death cannot be excluded. In this context, SARS-CoV-2 antigens and genes have been detected in the AV-node of the cardiac conduction system [69]. This finding pointing to a direct infection explains the incident arrythmias and the conduction system disease observed in SARS-CoV-2 infection [69,70,71]. Also, other possible mechanisms include endothelial cell infection and endotheliitis, transcriptional alteration of multiple cell types in heart tissue, high levels of pro-inflammatory cytokines, activation of the transforming growth factor beta (TGF-β)/SMAD signaling, downregulation of ACE2 and dysregulation of the RAS system [67,72,73]. Some of these mechanisms are favored by ACE2 expression in the human heart, leading to heart injury among patients infected with SARS-CoV-2 [74]. The hyperactivated immune response has been also suggested as a putative cause of the cardiovascular long COVID complaints [67,72,75]. Another hypothesis suggests an incorporation of the SARS-CoV-2 genome in the genetic material of infected human cells, which might be expressed as chimeric transcripts, resulting in continuous activation of the immune-inflammatory-procoagulant cascade [76,77]. Consequently, the proposed pathways could explain the persistence/recurrence of cardiovascular complications following SARS-CoV-2 infection representing part of the symptoms of long COVID. 

## 5. RAS in Respiratory System, the Main Target of SARS-CoV-2, Therefore, Long COVID

Nearly 30% of patients with long COVID, including those with moderate illness, show respiratory symptoms such as cough and dyspnea [6,78,79]. Some of those infected may not fully recover, resulting in long-term complications. This is not surprising, since ACE2 receptors are abundantly present on the surface of the lungs. Spike (S) proteins on the SARS-CoV-2 virus attack these receptors, facilitating its entry into human cells, gaining control of the cell, and using it for replication. This impairs the enzymatic action of the ACE2 receptors, leading to increased lung symptoms due to RAS impairment [80].

Long-term breathing problems can be caused by the scarring of lung tissues, known as fibrosis. The latter impairs gas exchange in the lungs by reducing lung tissue flexibility [81]. These damages can persist for long periods after the disappearance of acute symptoms. For example, a study showed abnormalities in the lungs of COVID-19 patients with impaired oxygen uptake for up to nine months after leaving the hospital [82]. In agreement with these findings, CT imaging in COVID-19 patients showed air trapping for more than 200 days after they were diagnosed with acute COVID-19 infection. Regardless of the initial severity of infection, small airway disease occurred, revealing a significant incidence of long-term air trapping [78]. The persistence of respiratory problems over this period raises concerns about irreversible airway remodeling and fibrosis following SARS-CoV-2 infection, with long-term consequences that are still unknown. Despite the complications in the respiratory system, there is still a glimmer of hope for recovery. In fact, even after a severe case of COVID-19, infected patients’ lungs might recover with time, but the recovery from lung damage is a lengthy one. Indeed, the initial injury to the lungs is followed by scarring, and the tissues eventually heal. However, restoring a patient’s lung function to pre-COVID-19 levels might take anywhere from three to 12 months [83].

## 6. Long COVID and RAS in Immune System 

A cohort analysis of long COVID patients followed systemically for eight months after SARS-CoV-2 infection highlighted the long-term complications impacting the immune system [84,85,86]. After four months of infection, six proinflammatory cytokines (IFN-β, IFN-λ1, IFN-γ, CXCL9, CXCL10, IL-8 and soluble T cell immunoglobulin mucin domain 3 (sTIM-3)) were significantly raised in the long COVID patients and the asymptomatic matched control (MC) groups compared to those infected with other human coronaviruses (HCoV) and unexposed healthy control (UHC) groups. Type I IFN (IFN-β) and type III IFN (IFN-λ1) remained highly expressed in the long COVID group even after eight months of the infection. These patients had a persistent activation of innate immune cells (myeloid and plasmacytoid dendritic cells (pDC)) and lacked naive B and T cells [85]. These findings highlight the persistence of the pro-inflammatory phenotype in long COVID patients that could trigger subsequent complications. Moreover, plasma ACE2 activity was shown to be increased in long COVID patients [85,87]. In fact, plasma ACE2 activity was evaluated at months three, four, and eight after infection. The median plasma ACE2 activity was considerably higher in the long COVID and MC groups after three and four months of infection compared to the HCoV group. However, plasma ACE2 activity in the LC and MC groups decreased to become comparable to those of the HCoV and UHC groups by month eight. Together, these findings demonstrate that the plasma ACE2 activity parameter is implicated in SARS-CoV-2 infection and is not a common property of other coronaviruses [85]. 

Mast cell activation (MCA) has also been described as a possible mediator of the hyper-inflammatory responses found in acute COVID-19 infection and long COVID [88,89]. In fact, uncontrolled chemical mediator release causes a wide range of symptoms in MCA syndrome (MCAS), many of which are also experienced by long COVID patients [90,91,92]. Symptoms of MCA were increased in long COVID patients, along with a degree of severity that is similar to MCAS [93]. 

Several hypotheses have been proposed to explain the promotion of MCA in long COVID:
(i).Intricate connections between stressor-induced cytokine storms and epigenetic variant-induced states of genomic fragility trigger further somatic mutations in stem cells or other mast cell progenitors [91].(ii).Mast cell and microglia activation by cytokines or the SARS-CoV-2, possibly via the RAS [89,94].(iii).SARS-CoV-2-induced gene dysregulation, resulting in mast cell genetic regulatory loss [95].(iv).Autoantibodies reacting with immunoglobulin receptors on mast cells [96].(v).SARS-CoV-2-induced increase in Toll-like receptor activity leading to mast cell activation [97]. 

Clearly, COVID-19-induced inflammation is multifaceted and additional immunological problems seen in acute infection, such as excessive macrophage dysfunction and platelet serotonin release, highlight the impairment of the immune system homeostasis. The persistence of this impairment may play a role in long COVID complications facing different organs. 

## 7. The Expression of RAS in the Urinary System Extends Long COVID Symptoms 

Patients with COVID-19 may also experience new onset or flare-ups of baseline urinary symptoms, which are less well-known but increasingly reported. Although the underlying pathophysiology of urine symptoms in COVID-19 patients is unknown, concepts have arisen from smaller, single-center investigations that explained COVID-19’s impact on the genitourinary system [98,99,100,101]. Nevertheless, whether urinary symptoms and any accompanying problems exist in protracted COVID or post-acute COVID-19 syndrome, patients need further exploration. Survey-based results show new or worsening overactive bladder symptoms (OAB) months after SARS-CoV-2 infection. In fact, 71% of participants experienced new urine symptoms following the infection, while 29% of patients who had previously experienced OAB reported worsened manifestations after SARS-CoV-2 infection. Since this study was conducted on hospitalized subjects, more investigations are needed to check for urinary symptoms in patients who had asymptomatic, mild, or moderate infection [102]. 

As for the potential mechanism of action, ACE2 receptors are found in bladder urothelial cells [100,103]. Therefore, these cells are vulnerable to SARS-CoV-2 infection and subsequent dysregulation. As a result, it’s possible that the de novo or exacerbated OAB symptoms seen in the proposed investigation are a side effect of a cellular cascade triggered by ACE2 receptor activation in the bladder. In agreement with this hypothesis, the presence of SARS-CoV-2 in the urinary tract has been validated, and the virus was also detected in urine [104]. Although this observation was not common to all patients, it highlights the extended route of the virus in the body [105]. Other studies also suggested a viral cystitis associated with COVID-19 [98,101]. While causation cannot be confirmed, a hypothesis can be proposed. For example, COVID-19 participants with new onset of severe urine symptoms had higher levels of pro-inflammatory cytokines, suggesting that COVID-19-related inflammation could lead to bladder dysfunction [99]. 

In addition, COVID-19 is associated with an increased risk of post-acute sequelae involving renal systems, where ACE axis elements are expressed, and most circulating RAS proteins are filtered by the renal glomeruli before being reabsorbed by the proximal tubule [106,107]. When tissue RAS is dysregulated, as in hypertension, it causes considerable renal inflammatory damage. The vital anti-inflammatory role of AT2R and ACE2 components of RAS in the kidneys to maintain renal homeostasis then plays a part. However, the perturbation of ACE2 caused by SARS-CoV-2 could potentially explain the renal damage [108]. In a cohort of 89,216 COVID-19 survivors, the investigation of post-acute renal sequelae of SARS-CoV-2 showed an increased chance (and incidence) of acute kidney injury (AKI), end stage kidney disease (ESKD), major adverse kidney events (MAKE), and estimated glomerular filtration rate (eGFR) decrease after the first 30 days of infection [109]. 

Of note, the intensity of the acute infection raised the dangers of renal outcomes. For example, individuals who survived COVID-19 had a larger loss of eGFR than non-infected controls, and eGFR loss was more severe as the intensity of the acute SARS-CoV-2 infection increased [109]. These findings imply that persons with COVID-19 have a higher risk of negative renal outcomes after the acute phase of infection has passed. Once again, evidence shows that long term post-COVID sequels can target different systems and people with COVID-19 should receive post-acute care that includes attention and treatment for both acute and chronic kidney disease. In the post-acute phase of SARS-CoV-2 infection, the mechanism, or processes of increased risk of AKI, ESKD, MAKE, and eGFR decrease are still unidentified. Although preliminary findings suggested that the SARS-CoV-2 may exhibit kidney tropism, more recent evidence detected very few viral RNA reads in kidney tissue [110].

## 8. The Outcomes of Long COVID on Metabolic Disorders, a Focus on Diabetes

COVID-19 and type 2 diabetes mellitus (T2DM) have a bidirectional association. Diabetes with poor control worsens COVID-19 and is linked to an increased risk of morbidity and mortality. Unfortunately, the COVID-19 pandemic has also led to poor diabetes control, the evolution of prediabetes to diabetes, an increase in the number of new cases of diabetes, and an increase in corticosteroid-induced diabetes [111,112,113]. Although no significant link was observed between the susceptibility to SARS-CoV-2 infection and diabetes, the pathology enhances the severity, the fatality rates, and complications of COVID-19 [114,115]. These outcomes raise from hyperglycemia and blood glucose fluctuations, expression of furin proteins and ACE2 receptors, ACE2 autoantibodies generation, immunological and inflammatory system imbalances, diabetes-related comorbidities, and lung injury in diabetes [116]. In human monocytes, increased plasma glucose levels and glycolysis accelerate and maintain SARS-CoV-2 replication [117], whereas increased plasma glucose levels enhance viral replication [118]. These findings suggest that hyperglycemia may impair antiviral immune responses against SARS-CoV-2 infection, which might explain why diabetic individuals with COVID-19 have a longer recovery period and more severe COVID-19 symptoms [119].

As previously mentioned, the mechanism in which diabetes may increase the risk of serious infection can be due to enhanced expressions of the ACE2 receptor and furin, which may facilitate SARS-CoV-2 entrance and replication [120]. In fact, SARS-CoV-2 binds to ACE2, which can be then cleaved by proteases like TMPRSS2 and furin, resulting in virosome complex internalization [121]. Therefore, the elevated levels of serum furin, a sign of diabetes progression linked to metabolic disorders and a higher risk of diabetes-related death, favors the entrance of SARS-CoV-2. Elevated fibrinolytic enzyme levels were also linked to diabetes. These proteases can break the S protein of SARS-CoV2, allowing the virus to interact with ACE2 and enter cells, hence increasing the virus’s virulence and infectivity [122]. Together these findings suggest that the COVID-19 link is sponsored by RAS and its player ACE2, even in metabolic diseases. 

The post-COVID-19 period may be complicated for diabetic people [123]. In fact, the infection may aggravate diabetes complications and raise the risk of death by causing sodium/glucose cotransporter 1 (SGLT1) imbalance in the intestinal epithelium that is mediated by ACE2 [124,125,126]. The high levels of ACE2 in intestinal mucosal cells as well as the gallbladder make these organs possible sites for viral entry and replication [127]. Regarding the mechanism of action, it has been hypothesized that SARS-CoV-2 downregulates ACE2, resulting in increased Ang II and a concomitant reduction in Ang-(1-7) levels. Since Ang II up-regulates SGLT1 and Ang-(1-7) suppresses it, the infection results in SGLT1 upregulation, enhancing intestinal glucose absorption, and facilitating the development of hyperglycemia in COVID-19 patients [128]. 

In the same context, highlighting the involvement of RAS in linking diabetes and COVID-19, the liver, endocrine pancreas, adipose tissues, kidneys, and small intestines are all rich in ACE2 [124,127]. As a result, it is likely that SARS-CoV-2 could bind ACE2 in key organs as the liver and the pancreas, causing cell destruction and insulin resistance, and therefore aggravating the diabetes prognosis or causing new-onset diabetes. In agreement with this hypothesis, a new-onset diabetes diagnosis was shown to be prevalent in SARS-CoV-2 infections as part of long COVID syndrome and is not linked to a preexisting history of diabetes or glucocorticoid usage. After hospitalization, roughly 14% of COVID-19 patients acquire new-onset diabetes, according to an analysis conducted on more than 3500 patients [129]. Furthermore, COVID-19 patients show insulin resistance, a symptom of type 2 diabetes [129].

During the COVID-19 pandemic, there was also evidence of an increase in pediatric type 1 diabetes [130,131], and an increase in the prevalence of diabetic ketoacidosis (DKA) and severe ketoacidosis at diabetes diagnosis [132]. However, the direct relationship between SARS-CoV-2 infection and an elevated probability of diabetes diagnosis is not clear yet. One suggested reason pointed to the possible fear of individuals to access health care [133,134]. Having seen the increased risk of diabetes upon individuals with a SARS-CoV-2 infection, it is important to keep an eye on blood glucose levels after the acute period. In fact, long-term uncontrolled diabetes can cause organ damage, especially microvascular destruction, which can be aggravated in patients infected with SARS-CoV-2. Diabetes also raises the likelihood of developing severe and critical COVID-19 illness, as well as the necessity for hospitalization and artificial ventilation [132,135].

On the other hand, corticosteroid therapy for COVID-19, which was commonly used in India, caused additional complications [136]. High-dose corticosteroid therapy results in several side effects including hyperglycemia, electrolyte imbalance, and myopathy. In general, steroids raise blood sugar levels through improving the action of counter-regulatory hormones on hepatic gluconeogenesis, or the generation of glucose from the liver. They also lead to increased insulin resistance by blocking insulin’s activity. Steroids can act on muscles and adipose tissue to lower glucose absorption and might directly impair the activity of Beta cells. Corticosteroids trigger inflammatory cytokines, such as those reported in COVID-19, and exacerbate insulin resistance [136]. These findings provide evidence that long COVID symptoms are not only the sequelae of SARS-CoV-2 infection, but they can also be triggered by medical practices. 

Of note, SARS-CoV-2 infection was shown to also affect other metabolic activities of the body. For example, antigens of SARS-CoV-2 were detected in the thyroid gland [137] and signs of thyroid disease were reported [138]. These findings could explain the fatigue symptom observed in long COVID patients, which is probably due to hypothyroidism. Another study investigated whether the persistent fatigue was due to vitamin D; however, results showed that persistent fatigue is independent of vitamin D [139]. Together, these studies emphasize yet again the pleiotropism of SARS-CoV-2 infection.

## 9. Long COVID-19: Follow Up and Potential Treatment 

As previously stated, long COVID syndrome is a multi-system illness characterized by respiratory, cardiovascular, hematologic, endocrine, and neuropsychiatric symptoms, which can occur solely or combined. As a result, therapy should be tailored to the individual and follow an interdisciplinary approach that addresses the clinical and the psychological complications. Coexisting illnesses such as diabetes, chronic renal disease, and hypertension should be treated as effectively as possible before they are worsened by the infection [5,140]. In general, a healthy lifestyle can also be proposed to help and reduce long COVID symptoms. Patients should be advised to follow a well-balanced diet, get enough sleep, limit their alcohol intake, and quit smoking [140].

In addition, organ-specific care should be given to patients after recovery from COVID-19. For example, patients who have chronic pulmonary symptoms may benefit from enrolling in a pulmonary rehabilitation program, which is critical for rapid clinical recovery. Contrarily to some findings showing that steroids worsen COVID-19 symptoms, a short investigation of COVID-19 patients four weeks after discharge found that early steroid administration resulted in rapid and considerable recovery [140]. Patients with prolonged heart symptoms after recovering from COVID-19 should be closely monitored by cardiac function tests such as electrocardiogram (ECG) and echocardiography to rule out arrhythmias, heart failure, and ischemic heart disease. In addition, given the increased risk of myocarditis in COVID-19 patients, a magnetic resonance imaging (MRI) of the heart may be undertaken if clinically appropriate to check for myocardial fibrosis or scarring. Even though COVID-19 is linked to a prothrombotic condition, there is presently no consensus on the benefits of venous thromboembolism (VTE) prevention in the outpatient environment. Current guidelines propose anticoagulant therapy for at least three months in patients with VTE in the setting of COVID-19 [141]. 

Patients should also be followed for potential common psychological disorders like anxiety, depression, sleeplessness, and PTSD, and if necessary, transferred to behavioral health professionals. Because of the wide variety of neurological symptoms associated with the condition, a neurology evaluation should be performed as soon as possible. The evaluation can be done together with laboratory analysis to investigate metabolic conditions known to worsen mood [142]. In case of suspicious seizures or paresthesias, electroencephalography and electromyography should be considered. Additional laboratory tests, such as hemoglobin A1C (HbA1c), thyroid-stimulating hormone (TSH), thiamine, folate, and vitamins, must be administered to evaluate for possible metabolic disorders [143].

In the absence of traditional risk factors for type 2 diabetes, serologic testing for type 1 diabetes-associated autoantibodies as well as repetitive post-prandial C-peptide measurements must be acquired at follow-up in patients with newly diagnosed diabetes mellitus, whereas it is appropriate to treat patients with such risk factors analogous to ketosis-prone type 2 diabetes [144].

The pathophysiology of long COVID is significantly linked to the RAS. Therefore, different therapeutic strategies involve drugs that can interplay in the RAS signaling pathway. For example, to counteract the virus-induced RAS disruption, ACE inhibitors that reduce Ang II synthesis were suggested [145]. This strategy modulating ACE activity is supposed to protect COVID-19 patients [146]. However, other enzymes, which are not sensitive to these chemicals, can take over the synthesis of Ang II and thereby limit therefore efficiency of the therapy [146,147].

Instead of preventing the generation of Ang II, an alternative technique could be to target AT1R. Angiotensin II receptor blockers (ARBs), a class of anti-hypertensive drugs that block the AT1R, have been proposed as potential pharmacological treatments to treat COVID-19-induced lung inflammation [148]. The selectivity of existing AT1R- blockers’ is advantageous, since it retains the protective effects of AT2R. These blockers are commonly used by people with hypertension to manage their blood pressure [149,150]. The abundant expression of these receptors in the heart, blood, arteries, kidneys, and intestine [151], suggest that inhibiting their activity lowers blood pressure and protects kidneys and the heart. Furthermore, inhibiting AT1R may attenuate acute lung damage and inflammatory mediator responses [152]. Taken together, it is reasonable that long COVID treatment focuses on RAS.

On the other hand, it has been shown that vitamin D is necessary for preventing the virus’s harmful effects, particularly in the brain. Since vitamin D reduces the generation of renin and the activation of the pro-renin receptor, it has a negative regulatory effect on the RAS. In fact, vitamin D, particularly D3, can effectively combat brain damage in persons with neurological illnesses by inhibiting the overactivation of systemic RAS, as well as brain RAS. Since there hasn't been enough research on long-term COVID-19 therapy, further information is needed to have a more detailed idea about the management of this disease [153,154]. 

## 10. Conclusions

It is obvious that SARS-CoV-2 attacks the organism after taking advantage of its tropism in different organs resulting in immune system dysfunction, increased inflammatory processes, and different other systems’ impairment, leading to long COVID (Figure 3). The primary culprit, however, is the virus’s persistence in the body after the acute phase. As such, the residual presence of the virus in some tissues is worrying. In long-term COVID patients, the virus seems to not be eliminated; instead, it gets reactivated, and its detrimental effects persist over time. In parallel, as previously stated, SARS-CoV-2 can infect different types of cells by attaching to their ACE2 receptor, including neurons, astrocytes, microglial cells, and oligodendrocytes. As such, the RAS system plays a key role in the virus’s pathogenesis as well as in the development of post-acute sequelae. The expression of ACE2 receptors in various organs explains the diversity of long COVID symptoms. Therefore, on one hand, it represents a major problem making different cell types vulnerable to SARS-CoV-2 infection; however, on the other hand, developing therapeutic strategies involving RAS can have beneficial effects to multiple organs. What is important is to manage long COVID symptoms before having severe complications leading to more complicated chronic diseases. 

## Figures and Tables

**Figure 1 molecules-27-02903-f001:**
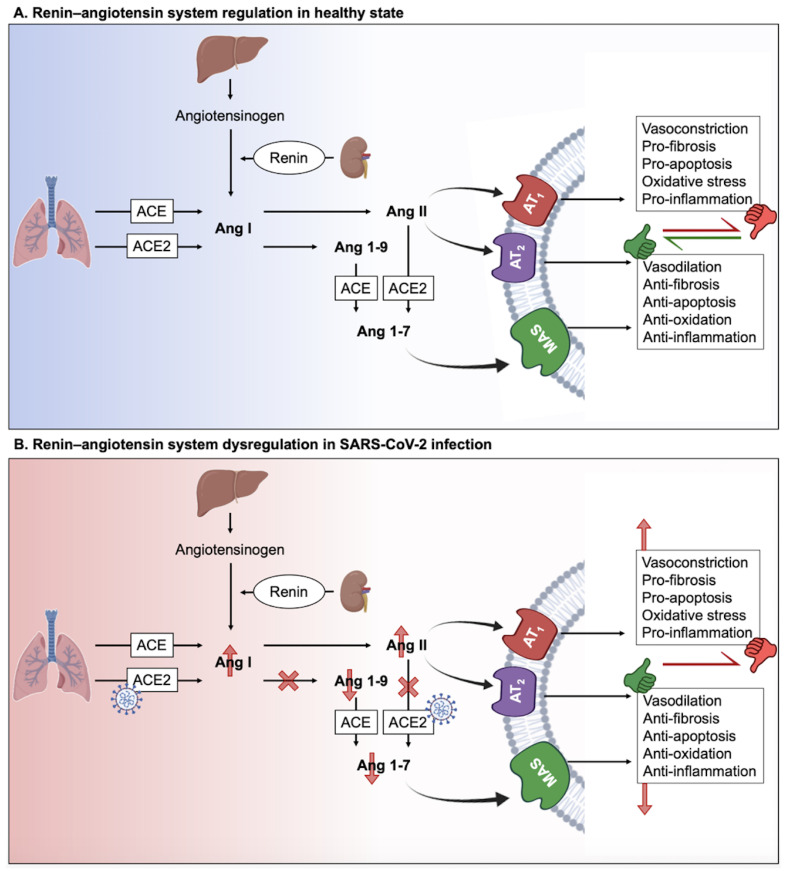
**Schematic of****RAS homeostasis (A) and impairment during****SARS-CoV-2 infection (B).** ACE: Angiotensin-converting enzyme; Ang: Angiotensin; AT1R: Angiotensin II receptor type-1; AT2R: Angiotensin II receptor type-2; MAS: Mas-related G protein-coupled receptors.

**Figure 2 molecules-27-02903-f002:**
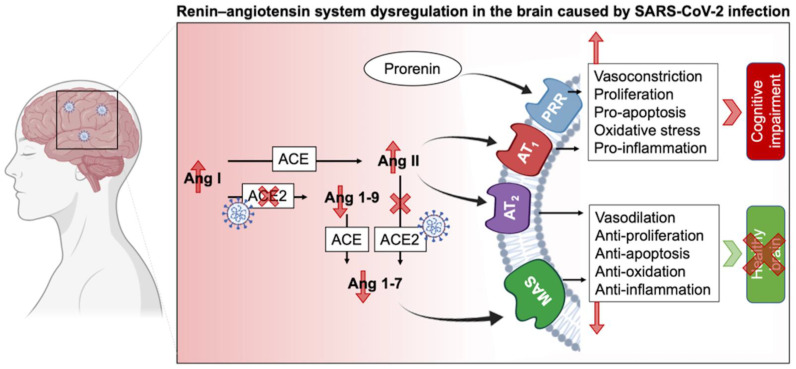
**Schematic of****RAS impairment in the brain during****SARS-CoV-2 infection.** ACE: Angiotensin-converting enzyme; Ang: Angiotensin; AT1R: Angiotensin II receptor type 1; AT2R: Angiotensin II receptor type 2; MAS: Mas-related G protein-coupled receptors, PRR: Prorenin receptor.

**Figure 3 molecules-27-02903-f003:**
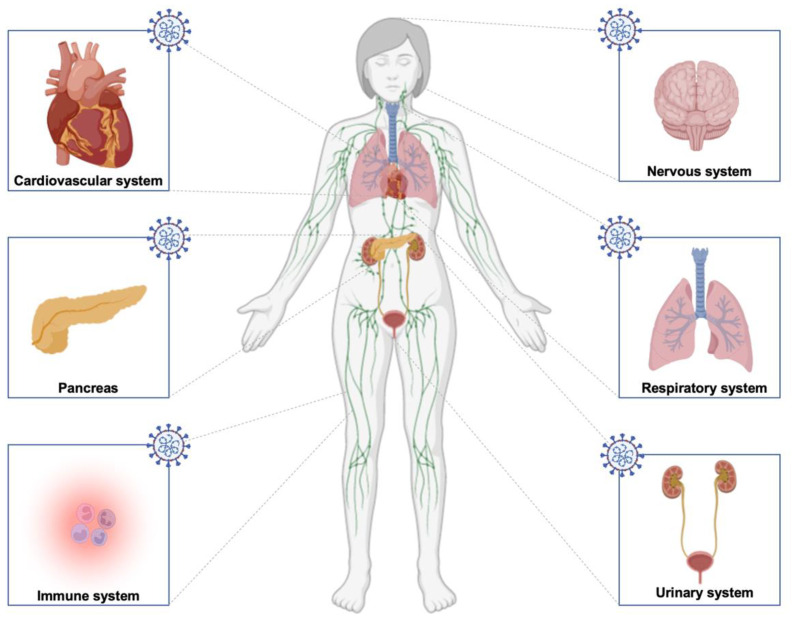
**Schematic of different systems and organs affected by long COVID.** The infection can impact the nervous, cardiovascular, urinary, immune, and respiratory systems and the pancreas in diabetes conditions.

**Table 1 molecules-27-02903-t001:** Summary of symptoms associated with long COVID and their related systems.

Symptoms	Systems
Fatigue	Nervous system
Attention deficit	
Sleep disturbances	
Cognitive impairment	
Post-traumatic stress disorder	
Muscle pain	
Concentration problems	
Headache	
Pins and needles sensation	
Anxiety	
Depression	
Delusions Paranoia	
Anosmia	
Ageusia	
Pericarditis	Cardiovascular system
Myocarditis	
Heart failure	
Thromboembolic illness	
Ischemic and non-ischemic heart disease	
Arrhythmias and palpitations	
Cerebrovascular disorders	
Eye disorders Skin lesions	
Cardiac abnormalities	
Cardiac arrest	
Sore throat	Respiratory system
Dyspnea	
Persistent cough	
Interstitial pneumonia	
Chest pain	
Activation of innate immune cells	Immune system
Inflammation	
Mast cell activation syndrome	
Macrophage activation syndrome	
Overactive bladder symptoms	Urinary system
Cystitis	
Renal damage	

## Data Availability

Not applicable.

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
