# Peer review of "The Pathophysiology of Long COVID throughout the Renin-Angiotensin System"

_molecules, 2022, doi:10.3390/molecules27092903_

Round 1
Reviewer 1 Report
This is an important topic and analyzing Long COVID in light of RAS is very appealing. Clearly, there is a depth of knowledge and expertise embedded in the manuscript by the authors. Nonetheless, I believe, there is a room for improvement for the essence of hypothesis to come across more clearly.
Major points:
1. Although RAS is abnormal during acute infection (as clearly described), it will be very helpful to include a paragraph for the current knowledge on the kinetics and time line for abnormal expression of ACE2 or other components of RAS following 3 or 12 weeks post infection.
2. Suggest including a clear statement that the manuscript involves the role of RAS primarily following a severe COVID-19 infection with high viral load and inflammation, and the mechanisms of Long COVID may differ after mild or asymptomatic infections.
Minor points
Section 3 (Long COVID in the nervous system): to my knowledge, there is lack of evidence for SARS-CoV-2 infecting neurons based on the post-mortem samples. Suggest editing section 3.1, line 18.
Section 3: May rearrange the paragraphs i.e. status of CNS --.PNS --> ANS, then role of endothelial damage --> role of inflammation --> role of autoantibodies (GBS) in Long COVID.
Section 2, last paragraph: Discuss inflammation and RAS more clearly. Is RAS related to remaining cell damage from cytokine storm? Please note levels of CRP and ESR can be normal in Long COVID.
Section 2, paragraph 2 - line 5: Angiotensin type-1 receptor should be type 2 (AT2). Also suggest consistency for using AT1R vs AT1 (line 4)
Author Response
This is an important topic and analyzing Long COVID in light of RAS is very appealing. Clearly, there is a depth of knowledge and expertise embedded in the manuscript by the authors. Nonetheless, I believe, there is a room for improvement for the essence of hypothesis to come across more clearly.
Thank you for your reviewing and comments. Please find below the answers for all your requests.
Major points:
- Although RAS is abnormal during acute infection (as clearly described), it will be very helpful to include a paragraph for the current knowledge on the kinetics and time line for abnormal expression of ACE2 or other components of RAS following 3 or 12 weeks post infection.
We couldn’t find a clear statement describing the kinetics of ACE2 or RAS during the infection. However, it is reported that ACE2 activities can predict the severity of mortality rates of the infection, as the activity was highly increased in the severe patients (see doi:10.1016/j.ijid.2021.11.028) this idea was used to put the statement you raised in comment nb2 in the context.
- Suggest including a clear statement that the manuscript involves the role of RAS primarily following a severe COVID-19 infection with high viral load and inflammation, and the mechanisms of Long COVID may differ after mild or asymptomatic infections.
The statement was added in Section 2 last paragraph
Minor points
Section 3 (Long COVID in the nervous system): to my knowledge, there is lack of evidence for SARS-CoV-2 infecting neurons based on the post-mortem samples. Suggest editing section 3.1, line 18.
The infection was shown to be possible using in vitro (doi:10.1080/22221751.2021.2024095) and mice models (doi.org/10.3390/pathogens11020257). The statement was updated accordingly.
Section 3: May rearrange the paragraphs i.e. status of CNS --.PNS --> ANS, then role of endothelial damage --> role of inflammation --> role of autoantibodies (GBS) in Long COVID.
We totally agree with the author’s comment. We separated the paragraphs and gave them titles to facilitate the reading.
Section 2, last paragraph: Discuss inflammation and RAS more clearly. Is RAS related to remaining cell damage from cytokine storm? Please note levels of CRP and ESR can be normal in Long COVID.
Yes, since the ACE/Ang II/AT1R axis is promoted in Covid infection, the cytokine storm induces cell damage. We thank the reviewer for the remark about CRP, the paragraph was updated accordingly.
Section 2, paragraph 2 - line 5: Angiotensin type-1 receptor should be type 2 (AT2). Also suggest consistency for using AT1R vs AT1 (line 4)
The consistency was checked and updated in the manuscript.
Reviewer 2 Report
This review deals with an important topic of the role of the renin-angiotensin system in the pathophysiology of long COVID. The authors have covered the topic in an organized and interesting way. The manuscript is well written, and only minor concerns need to be fixed to be fit for publication as follows:
- Title: remove "(RAS)."
- Keywords: replace (RAS, ACE2, AT1R, AT2R, Ang II-AT1R axis) with their full terms.
- Replace "Introduction to long COVID, symptoms, and risk factors" with "Introduction".
- Page 7, "A series of studies have…….. sequelae [35]" is a long paragraph. Please divide it and cite more references as you mentioned "series of studies," not only one reference.
- It is highly recommended to add other schematic figures representing the role of RAS in the immune complication of long COVID as many cytokines are implicated.
- Throughout the manuscript, no need to give abbreviations to the full term that have not been repeated in the manuscript. E.g., electroencephalography (EEG) and electromyography (EMG).
Author Response
This review deals with an important topic of the role of the renin-angiotensin system in the pathophysiology of long COVID. The authors have covered the topic in an organized and interesting way. The manuscript is well written, and only minor concerns need to be fixed to be fit for publication as follows:
Thank you for your reviewing and comments. Please find below the answers for all your requests.
- Title: remove "(RAS)."
Done
- Keywords: replace (RAS, ACE2, AT1R, AT2R, Ang II-AT1R axis) with their full terms.
Done
- Replace "Introduction to long COVID, symptoms, and risk factors" with "Introduction".
Done
- Page 7, "A series of studies have…….. sequelae [35]" is a long paragraph. Please divide it and cite more references as you mentioned "series of studies," not only one reference.
The sentence was divided and reformulated as demanded.
- It is highly recommended to add other schematic figures representing the role of RAS in the immune complication of long COVID as many cytokines are implicated.
Thank you for the suggestion. Regarding the figures, in the revue we decided to detail the mechanism of RAS homeostasis in Figure 1 and then we updated the figure to fit with RAS in the CNS, the major part of the revue. Since the effects of the signaling pathway are common between the systems (protective/detrimental effects), we didn’t want to repeat the same mechanism in every cell/system.
We also thought of creating schematic figure of covid and the immune response (see file attached), however, this figure doesn’t add much to the context of our revue. Thank you for your understanding.
- Throughout the manuscript, no need to give abbreviations to the full term that have not been repeated in the manuscript. E.g., electroencephalography (EEG) and electromyography (EMG).
Done
Reviewer 3 Report
Authors bring interesting and valuable review of the current insights on post-(long)-COVID-19 syndrome regarding RAS disturbance.
It would be valuable if section 2 (The interplay between long COVID and renin-angiotensin system (RAS)) included more detailed elaboration on ACE2 and Ang1-7 effects, in particular those being protective/anti-iflammtory (rather than this is mentioned in nervous system section). Also, it would be worthy to mention widespread ACE2 expression underlying Sars-oV-2 pleiotropism. (see and cite doi:10.1152/ajplung.00119.2020)
In the cardiovascular section, it would be beneficial to mention that SARS-Cov-2 was detected even in cardiac conduction system, and cardiac arrythmias surfaced as a common finding in post-COVID-19 patients (see and cite).
Considering long COVID and metabolic disorders, thyroid involvement should be discussed, since thyroid destruction was found in COVID patients (doi: 10.1530/ETJ-22-0005.), while fatigue and exhaustion are leading symptoms of both, long COVID and hypothyroidism. Are there any data linking vit D level and propensity to long COVID after infection?
Author Response
Authors bring interesting and valuable review of the current insights on post-(long)-COVID-19 syndrome regarding RAS disturbance.
Thank you for your reviewing and comments. Please find below the answers for all your requests.
It would be valuable if section 2 (The interplay between long COVID and renin-angiotensin system (RAS)) included more detailed elaboration on ACE2 and Ang1-7 effects, in particular those being protective/anti-iflammtory (rather than this is mentioned in nervous system section). Also, it would be worthy to mention widespread ACE2 expression underlying Sars-oV-2 pleiotropism. (see and cite doi:10.1152/ajplung.00119.2020)
Few lines were added to detail the effects on each receptor type. We thank the reviewer for this insightful comment.
In the cardiovascular section, it would be beneficial to mention that SARS-Cov-2 was detected even in cardiac conduction system, and cardiac arrythmias surfaced as a common finding in post-COVID-19 patients (see and cite).
We highlighted this idea in the section. We thank the reviewer for the insightful comment.
Considering long COVID and metabolic disorders, thyroid involvement should be discussed, since thyroid destruction was found in COVID patients (doi: 10.1530/ETJ-22-0005.), while fatigue and exhaustion are leading symptoms of both, long COVID and hypothyroidism. Are there any data linking vit D level and propensity to long COVID after infection?
Even though our focus was diabetes, we added few lines about thyroiditis and vitamin D to give the reader more perspectives. Thank you for the insightful comment.